# Plant-Based Dairy Alternatives: Consumers’ Perceptions, Motivations, and Barriers—Results from a Qualitative Study in Poland, Germany, and France

**DOI:** 10.3390/nu14102171

**Published:** 2022-05-23

**Authors:** Dominika Adamczyk, Diana Jaworska, Daria Affeltowicz, Dominika Maison

**Affiliations:** Faculty of Psychology, University of Warsaw, 00-183 Warsaw, Poland; diana.jaworska@psych.uw.edu.pl (D.J.); daria.affeltowicz@psych.uw.edu.pl (D.A.); dominika.maison@psych.uw.edu.pl (D.M.)

**Keywords:** plant-based dairy alternatives, qualitative research, consumer study

## Abstract

Plant-based dairy substitutes have been gaining popularity in recent years, but consumer perspective on these products is still relatively unexplored. The purpose of the study was to investigate the potential of plant-based dairy alternatives, including consumers’ motives and the barriers to embracing this food category. A qualitative study (24 focus groups, 154 respondents) was conducted in three countries: Poland, Germany, and France. The study allowed us to describe the reasons for using dairy substitutes (curiosity, health reasons, influence of others), their perceived advantages, and the barriers to their use. The study also showed that the role of dairy differs between the surveyed countries and is related to culinary traditions. As a result, attitudes towards and motives for using dairy substitutes differ in the different countries.

## 1. Introduction

Dietary habits in Europe are changing. Every year more and more people are deciding to restrict animal-based products in favour of plant-based foods [1]. This trend is reflected by the rising popularity of plant-based alternatives, not only to meat but also to dairy products. Numerous studies have investigated the nutritional values of plant-based alternatives (e.g., [2]). However, consumers’ perspectives on this, which entail people’s needs, expectations, and knowledge of plant substitutes, are still not well understood.

Plant-based dairy alternatives can be divided into five categories: cereal-based (oat and rice), legume-based (soy and pea), vegetable-based (potatoes), seed-based (flax and hemp), and nut-based (almond, cashew, and coconut) [3]. They also include alternatives to milk, yoghurt, cheese, ice cream, butter, and cream, which are made out of water extracts from plants [4]. Compared to dairy, plant-based products contain less of certain mineral elements, such as calcium, phosphorus, magnesium, potassium, and sodium, as well as vitamins such as vitamin D and B12 [2,3]. For this reason, plant-based dairy alternatives are often fortified to provide the maximum amount of nutrients [3]. Many healthcare professionals claim that, in general, dairy products have a higher nutritional value than plant-based alternatives; however, they also agree that the latter can be a part of a healthy diet [5].

In addition to the objective nutritional characteristics of plant-based dairy alternatives, consumers’ perspectives on this product category are crucial. A study on the relationship between the characteristics of dairy-alternative beverages and consumer willingness to pay showed that health-related nutritional attributes, such as calories, protein, fat, vitamin A, and vitamin D, are widely recognized by consumers and that they have a significant effect on their willingness to pay for these products [6]. The protein content turned was reported to be the most important nutritional attribute in terms of its significance for the price [6]. It seems that the health-related characteristics of non-dairy drinks are significant for U.S. consumers. However, a study on consumer preferences towards plant-based dairy beverages conducted in Spain showed that, apart from the price, the product’s flavour was the major driving factor contributing to the utility of plant beverages [7]. Along with the health, price, and taste characteristics, environmental issues may be another reason why people reach for plant-based dairy alternatives [8]. Additionally, plant-based dairy alternative production processes are considered more ethical for animal welfare reasons, which some consumers perceive as a crucial factor when making food choices [9]. It needs to be stressed, however, that the mentioned studies only concerned plant-based dairy beverages and no other dairy-alternative products, such as yoghurt. In light of the results, which show that consumers who reach for plant-based beverages do not necessarily want to reach for plant-based yoghurts [6], it is important to consider all the products in the range of plant-based dairy alternatives due to potential differences that may appear in a consumer’s perception of various products.

In order to better understand the barriers to and triggers of consumer use of plant-based dairy alternatives, one should also take the psychological factors into account that may shape people’s food choices. Research has shown that individual dietary choices reflect people’s values, beliefs, knowledge, and attitudes [10]. Yet the studies investigating these factors in the context of plant-based dairy alternative consumption are scarce. What has been established so far is that people who are resistant to innovations and who have less knowledge of nutrition are more likely to avoid dairy in their everyday diet and to reach for plant-based dairy beverages but not for plant-based yoghurts [11]. Despite the fact that some conclusions on consumer perception of plant-based dairy alternatives can be drawn from previous research, this knowledge seems to be limited to one particular country (most of the studies were carried out in the U.S.) and to one type of product (the vast majority of the research only considered plant-based beverages). However, deeper consumer motives and needs in the context of plant-based dairy alternatives still remain little understood.

Another important question is whether the perception and level of acceptance of plant-based dairy substitutes is universal or whether it varies cross-culturally, for example, across different culinary traditions. According to some researchers, cultural factors are essential in shaping people’s dietary preferences, and they should not be overlooked in scientific research [12]. Although there are some studies conducted on this matter in New Zealand and the United States [13], studies investigating this issue in Europe are lacking. European countries differ in terms of food culture and traditions [14], and research should account for potential cross-country discrepancies, especially considering the data on dairy consumption, which shows differences in dairy intake in older populations of European countries [15]. Research has shown that the elderly populations of Poland, Germany, and France exhibit different dairy intakes, with the French consuming the most, and Poles the least dairy [15]. It is crucial to investigate the consumer perception of plant-based dairy alternatives in countries that differ in terms of their dairy intake.

The purpose of the current study was to investigate the potential of plant-based dairy alternatives—what are consumers’ motives and barriers to embracing this food category. Firstly, we aimed to explore the needs, expectations, and current consumer knowledge about a broader range of plant-based dairy alternatives (not only beverages). We decided to investigate this issue qualitatively, by conducting focus group interviews (FGIs) with consumers. A qualitative methodology provides a deeper understanding of the issues that may partly be unconscious and difficult to verbalize (e.g., barriers to acceptance of plant-based dairy alternatives). Moreover, a qualitative methodology is the usual manner of approaching new topics and, given that the knowledge of consumer perception of plant-based dairy alternatives is not well investigated, our aim was to explore this issue rather than to provide quantitative data on the usage and attitudes of this product category. Secondly, we aimed to compare consumers from three European countries: Poland, France, and Germany, which differ in terms of their food culture and traditions as well as the role of dairy in their diets. Cultural factors are crucial in shaping people’s dietary preferences and food choices. To the best of our knowledge, this is the first study to examine the differences in the perceptions of plant-based dairy alternatives between three European countries: Poland, Germany, and France.

## 2. Materials and Methods

### 2.1. Methods and Procedure

Focus Group Interviews (FGIs) were carried out in three countries: Poland, Germany, and France, in the second part of 2021. FGIs constitute a qualitative method in which a moderator discusses a topic with a group, noting the views of its individual members. The group participants exchange their insights and experiences with each other, and the group discussion provokes opinions and clashes of views [16]. This type of method is often used in marketing research when testing new products before launching them on the market but also in scientific consumer research [17].

A total of 8 FGIs were conducted in Poland (48 participants), 10 FGIs in Germany (52 participants), and 10 FGIs in France (54 participants). A total of 154 respondents participated in the study consisting of 24 FGIs. Due to the COVID-19 pandemic, all of the interviews were conducted online, on the Zoom platform. Each interview lasted approximately 2 h and was carried out in the native language of the participants.

All the interviews were carried out by moderators specializing in scientific social research and consumer research. A discussion guide was created that included topics on attitudes towards food in general, motives for reducing meat and other products of animal origin (dairy), and the perceptions on dairy and plant-based dairy alternatives. The interview was semi-structured in nature; hence, the moderator used the discussion guide while having the flexibility to elaborate on important themes that emerged during the interview. Since the study is part of a larger project (EIT Food, Dairy Alternative Drinks & Yoghurt based on rapeseed ingredients—DADYGo), the last part of the interview involved discussing the concepts for a new product, that is, a rapeseed-based yoghurt alternative. This part of the interview was not included in the analyses. The sessions were audio-recorded and transcribed verbatim.

The study was approved by the Ethics Committee of the Robert Zajonc Institute of Social Studies (number 5/21).

### 2.2. Participants

Recruitment was conducted by a recruitment agency specializing in social studies.

As required by qualitative research, the selection of respondents for the study was purposive and focused on attitudes toward nutrition and physical activity. The demographic variables were additive and had a control function. In each country, the interviews were conducted with three groups of respondents: (a) people focused on the diet (4 FGIs: 2 men, 2 women, and 2 younger (25–45) and 2 older (55+)); (b) people focused on the diet but also physically active in sports (2 FGIs: 1 FGI with women, and 1 FGI with men, ages 25–45); and (c) mothers of young children responsible for nutrition in the family and paying attention to diet (2 FGIs: 1 FGI with mothers of children aged 1–6 years; 1 FGI with mothers of children aged 7–12 years) (see Table 1). Such a selection of groups allowed for an exploration of the reactions towards plant-based dairy substitutes in groups of people with potentially different nutritional needs resulting from lifestyle and family life cycle.

All of the participants were consumers who make their own nutritional decisions (as well as decisions concerning children in the case of a group of mothers) and are interested in nutritional topics (new products, new diets, healthy nutrition, etc.), open to new products, looking for new nutritional solutions, ready to change their diet, and open to reducing animal products in their diet (or already reduced such products in their/family’s diet) as flexitarians (even if they do not define themselves in this way). The rejection criterion was the assessment of one’s financial situation as insufficient to meet everyday needs (people agreeing with the statements “There is not enough money even for the most urgent needs” or “We have to deny ourselves many things to have enough money to live on” were excluded from the research).

### 2.3. Data Analysis

The transcriptions of the interviews carried out in Germany and France were translated into Polish by a professional translator and then analysed using the MAXQDA 2022 qualitative analysis software [18]. A thematic analysis approach was applied to extract emerging major themes [19]. First, one of the researchers performed the process of coding the transcriptions. The extracted codes were then combined into main themes. This was followed by a process of organizing the codes and themes, in consultation with another researcher involved in the analyses. A total of 886 statement excerpts were coded, creating 89 codes from three main areas: an approach to eating (in general), attitudes toward dairy, and attitudes toward plant-based dairy alternatives. The codes in each area were combined into major themes. For the approach to eating, it involved the factors that are important in food and the reasons for limiting meat consumption; for dairy—the advantages and disadvantages of dairy products and, for plant-based dairy alternatives—the reasons for reaching for substitutes, the awareness thereof, and the advantages and disadvantages of such substitutes. The full code tree can be found in Appendix A.

## 3. Results

### 3.1. The Approach to Eating

#### 3.1.1. What Is Important in Food

For consumers, taste was the most important aspect of eating, as the food was one of the everyday pleasures of respondents. Some even described themselves as hedonists, underlining the importance of the taste sensations of meals. Treating taste as a very important dimension of food meant that the consumers in the study were not ready to eat tastelessly (sacrifice taste), even if something was very healthy. The taste was very often connected with seasonality, especially in the case of vegetables and fruits; choosing products that are in season makes the food tastier and heightens pleasure.

Health was also a very important dimension of nutrition, often being the motivator of changes in the diet. Healthy eating was, for many consumers, connected with looking for high quality food, which meant both checking the product ingredients (studying the label) and shopping in “good” places (e.g., buying meat at the butcher, not in the supermarket). Consumers declared that they gave up selected products (or reduced their amount in the diet) due to their bad health effects or, conversely, tried to introduce healthier products. Moreover, in the opinion of respondents, a healthy diet should be nutrient-rich and varied.

The price was also an important but not decisive factor in food selection. For the surveyed group of consumers, the value for money mattered more than the price itself; they were willing to pay more for better quality products (healthier, local, etc.). However, we have to remember that one of the respondent selection criteria was not being a price-driven consumer.

Some people focused strongly on buying local Polish/French/German products. They felt that the food produced in their country of residence was healthier (believing that their bodies were adapted to local products); hence, they supported local farmers. The biggest manifestation of “food locality” was the place where they shop. Local farmers markets were eagerly chosen for this purpose, or if purchases were made in chain stores, the country of origin was checked. Paying attention to locality, organic, or bio products had more to do with the health benefits for humans (bio = healthy and less processed).

The least important dimension of food was its sustainability (not harmful to the environment). This food choice criterion was very rarely spontaneously indicated by respondents. Moreover, the problem with sustainable food is that it is often more expensive than standard products. This makes it more difficult to “switch” to more sustainable products without spending a larger budget on eating or changing nutritional priorities.

For me, eating is a pleasure in the first place. Secondly, our health depends on it—it is said: “you are what you eat”. Over the past two years, I have changed my diet. I started paying more attention to what I eat. I focused more on the nutritional value of products. And I notice that it has a positive effect on my body.[woman, 25–45 years old, focused on diet, Germany]

#### 3.1.2. Differences between Countries

For German and French consumers, buying domestic products was connected with sustainability and considering the environment (due to a shorter supply chain and lower CO_2_ emissions). Some consumers seemed to strongly believe in the importance of a pro-ecological approach, while others were convinced that declaring such an attitude is related to political correctness; this was especially noticeable among German consumers. Polish consumers failed to spontaneously mention the importance of sustainability in the context of food in general.

#### 3.1.3. Reasons for Limiting Meat

The most important reason for limiting meat was related to health. Meat did not have benefits for some of the respondents; they felt unwell (“heavy”) after consuming meat and suffered from digestive system ailments. Some consumers noticed changes in their nutritional preferences; they no longer like the taste of meat and feel that they do not enjoy eating it. This is connected with two reasons: a change in the quality of meat (hence, its taste), resulting from mass production, and a change in the taste preferences of consumers resulting from aging processes (older respondents noticed that their taste perceptions have changed and that they now like different products than they used to). Some respondents believe that it is easier for them to stay in shape or lose weight by eliminating or limiting meat in their daily diet. Overall, an environmental motivation was the least frequently mentioned for discontinuing or reducing meat consumption.

It is also said that too much protein is not good for your health. Everything should be eaten in the right doses and in a balanced way. Additionally, we, as consumers, would like the slaughter to be performed with full respect. It is intended for nutrition, but not at all costs.[woman, 25–45 years old, focused on diet and daily exercise, France]

#### 3.1.4. Differences between Countries

The ecological aspect of meat reduction was mentioned much more often in Germany and France than in Poland. This is probably mainly the result of many years of education in the field of responsible consumer behaviour and the presence of the topic of sustainable food in the social discourse in these countries. Those who reduced meat consumption for ethical reasons also mentioned that they had decided to stop eating or to reduce the amount of meat consumed after watching reports from farms and slaughterhouses.

### 3.2. Dairy Perception

#### 3.2.1. Dairy Advantages

Dairy products were perceived as a very important part of the daily diet of the surveyed consumers. They were perceived as a very healthy category of products and a source of protein, calcium, and probiotics (building immunity and strengthening bones and muscles). The other mentioned advantages of dairy were taste and the category variety (alongside fruit and vegetables, dairy products are one of the most diverse food categories). The role of dairy products also appeared to increase as consumers restricted meat in their diets—even becoming the most important source of protein. A coding tree visualizing the network of associations with dairy is presented in Figure 1.

Besides, dairy products are very diverse. They provide a varied diet. You can adjust the fat content. If you prefer lean dairy or you don’t want to eat fat, you can choose one that has less fat. And if someone prefers more fatty products, they can choose them. We need the protein contained in cheeses and yogurts because we need calcium for bones, so we should eat it.[woman, 55+ years old, focused on diet, Poland]

#### 3.2.2. Dairy Disadvantages

Although consumers did mention the advantages of dairy products, some disadvantages were indicated during the discussion. They were mostly health-related. However, the perceived disadvantages were not associated with the dairy category in general but with individual dairy products. Processed dairy products, such as yoghurt with added sugar, were also a problem due to their high calorie content, preservatives, and flavour enhancers. Many doubts were raised about milk, as a product that contains harmful enzymes and causes digestive problems and allergies.

Some consumers did note dairy products as being unnatural for humans; they mentioned that cows’ milk is intended for calves, not humans. The respondents said that the habit of drinking milk is strange and that it is something that is ingrained in us from a young age but that it does not make any sense. They drew attention to the fact that a human is the only being who drinks milk all their life—other animals consume it only up to a certain point, after which they stop. However, these objections were not ethical in nature, but rather pointed out that adult mammals do not need to drink milk.

Children don’t necessarily need milk either. You can compensate for this in other ways. You have to have some fun with it. There are enough things to compensate for it. Since children have forever consumed cow’s milk, just like adults, however, there is not a single species, mammal, that also feeds on the milk of other mammals over time. We belong to mammals, just like a cow, like a mouse, like an elephant, and no other race of mammalian subgroups uses the milk of another. Why should we rely on it?[woman, 55+ years old, focused on diet, Germany]

The ethical issue of dairy consumption related to animal welfare did not arise spontaneously. Some individuals raised the issue as an additional argument for limiting the consumption of dairy products but not as a main reason for excluding them. However, these arguments relate only to factory farming where animals are mistreated, and not to the consumption of dairy products in general. Respondents believed that consuming dairy from a proven, local source did not raise ethical concerns. In the case of dairy products, the link between the animal and the milk was weaker than in the case of meat. On a rational level, consumers know that milk comes from animals, but they do not seem to think about it on a daily basis. Additionally, since it does not involve animal slaughter, eating dairy products does not feel “unpleasant”, as in the case of meat.

#### 3.2.3. Differences between Countries

The perception of dairy products as healthy food was more common among Polish consumers than among German or French buyers. In Germany and France, vegetables and fruits were prototypical healthy categories, whereas in Poland, both vegetables and dairy products were classified as the healthiest product groups. The differences, in this case, are due to the dairy eating traditions of the studied countries. For Poles, there is a clear distinction between milk and other dairy products. The prototypical healthy products are those containing healthy bacteria (yoghurt and kefir). Other dairy products traditionally eaten in Poland are also lean and low-calorie products, such as traditional Polish cheese, known simply as “white cheese”, and cottage cheese. These dairy products are perceived as low in fat and healthy; hence, the association with the dairy category is, in general, better. Whereas in France, for example, the prototypical dairy product is cheese (Poland does not have such a strong cheese-eating tradition), which, by its nature, is usually fatty and has a high calorific value. Thus, in France, there is a conflict between the strong rooting of cheese in the culinary tradition and its perception as extremely tasty on the one hand, and an awareness of its high caloric content on the other. In Poland, this problem does not exist—dairy products that are strongly rooted in the culture are, at the same time, relatively low in calories. Among German respondents, there were frequent statements about the harmfulness of dairy products to adults and the associated digestive difficulties. This was due to the fact that, for Germans, the prototypical dairy product is milk.

### 3.3. Plant-Based Dairy Alternatives

#### 3.3.1. Plant-Based Dairy Alternative Awareness

The knowledge of dairy alternatives varies considerably in the surveyed group of consumers, from people who choose different options on a daily basis, to people who have tried a few selected products, to those who have seen alternative products in the store but have not reached for them. In general, consumers were more familiar with products that resemble milk (soy milk, oat milk, coconut milk, almond milk, and rice milk). Among those less frequently mentioned, usually only by highly engaged consumers who are actively looking for this type of product on a daily basis, were other milk alternatives (pea milk, spelt milk, quinoa milk, millet groat milk, and hemp milk), yoghurts, butter, and cheese. The coding tree visualizing the network of associations with plant-based dairy alternatives is presented in Figure 2.

#### 3.3.2. Plant-Based Dairy Alternative Advantages and Disadvantages

Respondents did not note many advantages of dairy alternatives other than being a suitable product for people who are lactose intolerant. Most of them could not point out the visible advantages for themselves. Occasionally, and only among French consumers, there were comments about the lower caloric content of dairy alternatives compared to dairy products such as cheese. Some other respondents mentioned ecological and ethical benefits, but they were only an addition to the benefits for one’s health.

One of the most common associations with alternative dairy products is the high price of such products—much higher than traditional dairy products (according to respondents, two to four or even five times more expensive). In the case of some products, this price seems more justified—when the base ingredient like coconut or almonds, for instance, does not grow in the country and there is a need for it to be imported from more distant regions of the world. However, in the case of certain other products, especially products based on cereals like oats or millet, the price is perceived as unreasonably high and dictated mainly by the desire to profit from the sale of currently trendy products.

A large proportion of the surveyed consumers have tried selected alternative dairy products, but they often simply do not like them (these products do not match their preferences). People who tried these products as alternatives (to dairy) seem to be more dissatisfied with the taste than those who categorized them as a way to diversify their diet. The approach to the dairy alternative category as a ‘dairy substitute’ leads to it being perceived as inferior (the products pretend to be something they are not). Since there were no people in the study group who had to give up dairy products for health reasons, the question arises about the purpose of changing one category (dairy products) to another (alternative dairy products).

In my opinion, coffee does not taste good with dairy substitutes—but then you can drink black coffee. It’s the same with cheese. I buy animal cheeses because they taste better—cheese made from substitute products is unpalatable—it’s just too dry… It lacks fat, which is obtained from cow’s milk. Soy milk simply doesn’t have enough fat to make it taste good.[woman, 25–45 years old, focused on diet and daily exercise, Germany]

#### 3.3.3. Motives for Reaching for Plant-Based Dairy Alternatives

The main motivator for reaching for alternative dairy products is curiosity, especially concerning their taste. Most people who tried dairy alternatives are not looking for a taste similar to standard dairy products but something they will simply enjoy eating. Consumers who are more focused on diversifying their diets (both to gain a wider spectrum of flavours and to provide a variety of nutrients) treat dairy alternatives as an opportunity to expand their and their families’ diets. In the case of children, serving alternative dairy products is often associated with learning to be open to new tastes (the belief that the more tastes a child learns and experiences in childhood, the easier it will be for them to accept new tastes in the future). Other reasons, beyond curiosity and the need for diversity, include health reasons. Malaise and digestive system ailments after consuming cow’s milk make some people look for alternative dairy products or use lactose-free products. Other consumers have tried alternative dairy products under the influence of a significant person to them who reaches for this type of product themselves (wives or partners in the case of men, and children and in the case of senior citizens).

#### 3.3.4. Differences between Countries

German and French consumers were able to list noticeably more dairy alternatives than Poles. This may result from both the availability of the products and an attachment to traditional products (which seems to be slightly greater in Poland). The low knowledge of the dairy alternative category in Poland is also evidenced by the mention of dairy products sourced from other animals than cows as examples of dairy alternatives (e.g., Halloumi cheese, goats’ cheese, sheep’s cheese, goats’ milk yogurt, goats’ milk powder, and ghee). Mixing the dairy and non-dairy categories may also indicate that cows’ milk–based products are the prototype of dairy products in the minds of consumers; hence, all “non-cow” products are viewed as alternatives.

Ecological and ethical aspects of reaching for alternative dairy products appear very rarely as an important motivator for changing eating habits; it was spontaneously mentioned by some consumers from Germany and France only.

The advantages of plant-based dairy products were almost exclusively mentioned by participants from France. They usually paid attention to health benefits. Dairy alternatives were seen to be less fatty and make a person feel lighter after consumption (when compared with the prototypical for French consumers dairy category of hard cheeses). Other respondents also mentioned ecological and ethical benefits, but they were only an addition to the health benefits.

## 4. Discussion and Conclusions

### 4.1. Discussion

The conducted study showed some insightful results regarding the attitudes towards dairy and dairy substitutes. An important result is that dairy is the prototypical healthy product that plays an extremely significant role in the daily diet, and most people cannot imagine giving up the entire category of dairy products. If people are willing to include alternative products in their diet, they do so mainly out of curiosity, a willingness to experiment and try new flavours, and wanting to diversify their diet. However, they do not plan on giving dairy products up completely in favour of plant-based alternatives. It rather comes from curiosity and wanting variety, or a response to the adverse health effects of drinking milk. Such a perception of dairy may be due to numerous marketing campaigns aiming to promote dairy consumption that were conducted in various countries across the world; in Poland, a milk-promoting campaign was one of the most effective marketing endeavours in this country [20]. This may be one of the reasons why the perception of the healthiness of dairy was the strongest in Poland, compared to Germany and France. Moreover, research on U.S. participants has shown that people who consume more dairy in their childhood are more prone to have positive associations with this food category (compared to plant-based dairy alternatives) in their adult life, and they are more willing to give dairy to their children [21]. Therefore, such positive associations with dairy may result from both promotional and educational activities, as well as dietary habits.

Rarely is the motivation to give up dairy an ethical or environmental issue. When people are aware of the problems resulting from dairy consumption, they are pushed aside by hedonistic motives, for example, by the pleasant taste of dairy products. This fact is probably due to the weaker link between animal suffering and milk production, on the one hand, and the ecological aspects of industrial farming, on the other. While this connection seems already clear in the case of meat due to arguments in the media and in educational campaigns, it is not so obvious in the case of dairy alternatives. On a rational level, consumers know that milk comes from animals, but they do not seem to think about it on a daily basis. Additionally, since it does not involve animal slaughter, eating dairy products does not feel as “unpleasant” as in the case of meat. This result is interesting in light of previous research, which showed that when people are directly asked about the perception of the sustainability of dairy versus plant-based dairy alternatives, they indicate that the latter is better for the environment [8]. However, based on our study, it seems that the ethical and environmental arguments were not the most vital for consumers’ perspectives on plant-based dairy alternatives, as they rarely appeared spontaneously.

The study also showed that the role of dairy differs between the surveyed countries and is related to the culinary traditions present there. In Poland, for example, dairy is strongly perceived as a necessary and essential part of the diet and very good for health, especially yoghurt, kefir, and cottage cheese (due to them containing “good bacteria”). This approach is very rational and is linked to the knowledge of the health benefits of dairy products (which may partly be due to widespread social campaigns promoting milk and dairy consumption). The case is different in France, where dairy products are strongly associated with high-calorie cheeses but, at the same time, are considered an indispensable part of daily nutrition. In the case of the French, the attitude towards dairy is more emotional—they “love cheese”, and despite the fact that they are aware of its bad effects on health, they are not ready to give it up (conflict of the “heart and mind”). In the case of German consumers, dairy products are not so firmly rooted in food traditions. On the other hand, these consumers are somewhat more likely than the other two groups (although still relatively seldomly) to associate dairy products with sustainability, that is, linked to animal welfare and negative environmental consequences. The results showing the differences between these countries are in line with the report showing the discrepancies in food traditions and cultures as well as with the studies revealing the differences in dairy intake in France, Germany, and Poland [14,15].

The conducted study also shows that barriers to dairy alternatives still prevail (more rationally rooted in Poland and more emotionally in France) and, at the same time, the majority of consumers in the three surveyed countries see no real reason why they should replace dairy nor the advantages of dairy product alternatives. The Germans are slightly more open to sustainability issues in the context of dairy, but at the same time, sustainability in the context of this category (and also other food products) raises some doubts as to whether it is not politically motivated.

As we can see, in contrast to meat, where awareness of its harm to the environment and negative impact on animal welfare and consumer health is high, the dairy reduction is still a major challenge due to the many consumer barriers to alternatives, the deep grounding in dietary traditions, and the strongly held belief in the health benefits of this food category.

### 4.2. Limitations

The research findings, due to their qualitative and exploratory nature, should be treated with caution. Due to the nature of qualitative research, the sample in the study was purposive, so we do not have a complete picture of the different demographic groups. The next step to better understand perceptions of plant-based dairy alternatives would be to conduct a quantitative study showing how this topic looks across different demographic groups (in people of different ages, between urban and rural areas, etc.). The above-mentioned country differences should also be validated in a representative quantitative study. In the case of the awareness of plant-based dairy alternatives, an accurate determination of the level of awareness and testing (trial) of the individual types of products requires quantitative research on a representative sample.

In addition, the study sample was homogeneous in terms of income. The study only involved consumers whose income allowed them to satisfy their everyday needs, and some results (for example, the low importance of price in everyday food decisions) may be the result of such sampling. Additionally, the participants were people who are concerned with the importance of daily nutrition; hence, they perhaps have a strong focus on health in the daily diet. There may also be other groups of consumers who have different motives for reaching for substitutes, such as people who are more focused on ethical aspects.

## Figures and Tables

**Figure 1 nutrients-14-02171-f001:**
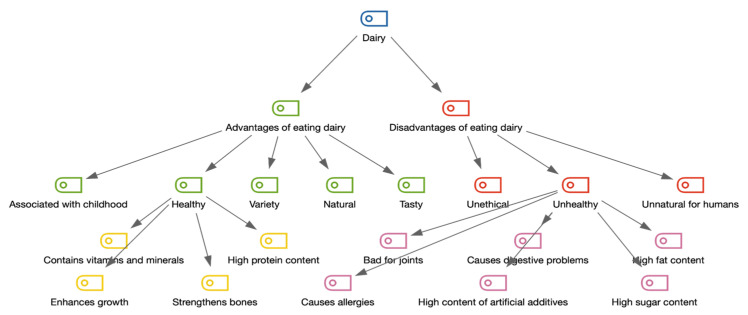
Visualization of codes connected to dairy consumption (network of associations regardless of frequency).

**Figure 2 nutrients-14-02171-f002:**
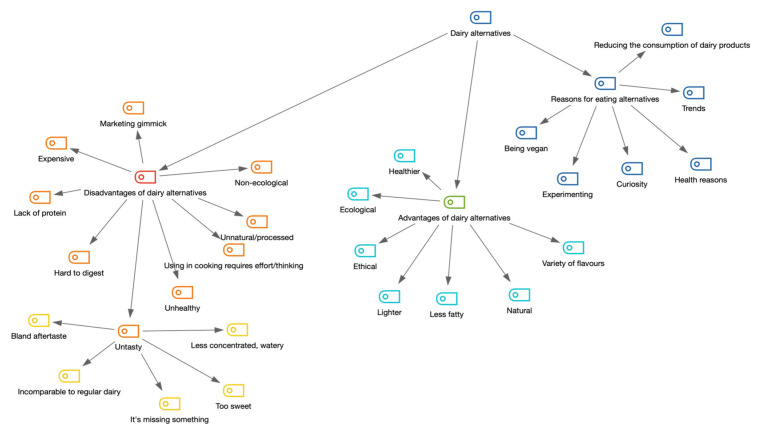
Visualization of codes connected to plant-based dairy alternative consumption (network of associations regardless of frequency).

**Table 1 nutrients-14-02171-t001:** Type of groups and number of FGIs and participants.

Type of Groups and Number of Groups			Number of Respondents in Each Country		
			Poland	Germany	France
Type 1	People focused on diet (diet concerned)	2 FGIs with women (1 FGI 25–45 years old and 1 FGI 55 + years old)	12 (6 + 6)	14 (9 + 5)	13 (7 + 6)
		2 FGIs with men (1 FGI 25–45 years old and 1 FGI 55 + years old)	12 (6 + 6)	12 (6 + 6)	13 (7 + 6)
Type 2	People focused on diet and physically active in sports	1 FGI women (25–45 years old)	6	7	8
		1 FGI men (25–45 years old)	6	8	6
Type 3	Mothers of young children responsible for family nutrition	1 FGI—mother with children 1–6 years old	6	6	8
		1 FGI—mother with children 6–12 years old	6	5	6
Total number of groups (16 FGIs)	8 FGIs	8 FGIs	8 FGIs	Total number of groups (16 FGIs)	8 FGIs
Total number of respondents (*n* = 154)	48	52	54	Total number of respondents (*n* = 154)	48

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
