# Peer review of "Plant-Based Dairy Alternatives: Consumers’ Perceptions, Motivations, and Barriers—Results from a Qualitative Study in Poland, Germany, and France"

_nutrients, 2022, doi:10.3390/nu14102171_

Round 1

Reviewer 1 Report

Dear authors, this study investigates the potential of plant-based dairy alternatives and what are consumers' motives and barriers to this food category. The topic of the manuscript is very interesting, presenting one of the current trends in diet and food choices, the use of plant-based dairy alternatives. 

The manuscript is well written, very explanatory and expresses the views and beliefs on diet of different consumers. However, some minor issues should be addressed.

  1. I suggest making the introduction shorter, keeping all the current information in a briefer way, which is easier for the reader to follow.
  2. For the first time, in line 201 is referred "that one of the respondent selection criteria was not being a price-driven consumer, and in line 489 'the sample was homogeneous in terms of income". I suggest all the selection and exclusion criteria of the participants should be addressed in detail at "2.2. participants" section.
  3. In connection with the previous comment, is there demographic data of urban and rural participants? If yes, please comment on their answers. If not, you may refer this at the "limitations". I suppose, you would draw interesting and useful information depending on participants' place of residence.
  4. Line 166. How did you ensure that the translation into Polish kept the original meaning? Is there a specific procedure that you followed?
  5. Lines 264-267. I guess it should be in italics. It seems as a participant's comment. 
  6. Lines 284-286 and 291-293 seem to repeat the same message. Please, keep it once. 
  7. Figure 1 and figure 2 are not connected to any paragraph of the main body of the text. Please indicate the paragraphs that they explain. 
  8. Also, for easiness of reading and understanding, the legend in figure 2 should have bigger font size.  

Reviewer 2 Report

I appreciate the project, its findings, and the paper. It's a nice addition to our understanding. My one suggestion is to tighten up the background section. As written, it feels a little bit just like a list of descriptive notes about study after study that isn't terrible cohesive and doesn't hold attention. Perhaps a graphic like a table can be used to summarize the main findings to date, and then significantly shorten up the text.

Very minor point - search for the word "diary" and then correct that misspelling where it exists.
